# Microstructure and Magnetic Properties of Grain Boundary Insulated Fe/Mn_0.5_Zn_0.5_Fe_2_O_4_ Soft Magnetic Composites

**DOI:** 10.3390/ma15051859

**Published:** 2022-03-02

**Authors:** Liang Yan, Biao Yan, Lei Peng

**Affiliations:** 1School of Intelligent Manufacturing and Control Engineering, Shanghai Polytechnic University, Shanghai 201209, China; yanliang@sspu.edu.cn; 2School of Material Science and Engineering, Tongji University, Shanghai 201209, China; 84016@tongji.edu.cn; 3Faculty of Electronics and Information Engineering, Jing Gang Shan University, Ji’an 343009, China

**Keywords:** spark plasma sintering, mechanical ball milling, soft magnetic composites (SMCs), grain boundary insulation, magnetic properties

## Abstract

Mn_0.5_Zn_0.5_Fe_2_O_4_ nano-powder was coated on Fe microparticles by mechanical ball milling combined with high-temperature annealing. The effects of milling time on the particle size, phase structure and magnetic properties of core–shell powder were studied. Scanning electron microscopy (SEM), energy-dispersive spectroscopy and X-ray diffraction showed that the surface of the milled composite powder was composed of thin layers of uniform Mn_0.5_Zn_0.5_Fe_2_O_4_ insulating powder. SEM also revealed a cell structure of Fe particles, indicating that the Fe particles were well separated and isolated by the thin Mn_0.5_Zn_0.5_Fe_2_O_4_ layers. Then, Fe/Mn_0.5_Zn_0.5_Fe_2_O_4_ soft magnetic composites were prepared by spark plasma sintering. The amplitude permeability of Fe/Mn_0.5_Zn_0.5_Fe_2_O_4_ SMCs in the Fe/Mn_0.5_Zn_0.5_Fe_2_O_4_ soft magnetic composites was stable. The resistivity decreased with the increase in sintering temperature. The loss of the composite core was obviously less than that of the iron powder core. Hence, the preparation method of Mn_0.5_Zn_0.5_Fe_2_O_4_ insulating iron powder is promising for reducing core loss and improving the magnetic properties of soft magnetic composites.

## 1. Introduction

Soft magnetic composites (SMCs) refer to ferromagnetic particles coated with an insulating medium and prepared into bulk materials by powder metallurgy, expanding the working frequency of metal soft magnetic materials to the GHz level [1]. Compared with traditional metal alloys, SMCs not only maintain the excellent soft magnetic properties of the materials, but also reduce the high-frequency eddy current loss and improve the application frequency [2]. In recent years, SMCs have played the roles of energy coupling transfer and conversion in various devices [3,4]. With the intensification of energy shortage and environmental problems, reducing the loss of soft magnetic materials and improving the efficiency of the magnetic core are of great significance in saving energy and controlling environmental pollution.

Cladding plays a very important role in SMCs and directly affects the comprehensive properties of SMCs. However, SMCs with organic coating [5,6], organic–inorganic coating [7,8] or inorganic coating [9] have non-magnetic coating layers, which reduce the saturated magnetic induction strength of the soft magnetic materials and improve the coercivity and thereby hysteresis loss. Therefore, SMCs were selected in this study. At the same time, ferrite, an SMC with high resistivity, was used as the insulating medium to replace the non-magnetic coating in traditional SMCs. The concept of core–shell heterogeneity was introduced to construct a new type of Fe-based SMC with inter-particle insulation [10].

In recent years, mechanical ball milling has proven its ability to produce composite particle powder. Compared with the sol–gel, chemical reduction, thin-film deposition or nitriding methods, this technology can produce a large amount of materials quickly, efficiently and cheaply [11,12]. The particle size and component content of the constituent materials can be easily controlled by changing the ball milling parameters. In this study, Fe/Mn_0.5_Zn_0.5_Fe_2_O_4_ core–shell heterostructure composite powder was prepared by coating a nano-Mn_0.5_Zn_0.5_Fe_2_O_4_ insulating layer on the surface of Fe micro-powder through mechanical ball milling. We confirmed the practicability of the Mn_0.5_Zn_0.5_Fe_2_O_4_ nano-powder-coated Fe micro-powder, and studied the effect of grinding time on the structure and magnetic properties of the composite powder. We found that these composite powders have homogeneous Mn_0.5_Zn_0.5_Fe_2_O_4_ layers, which are capable of diminishing eddy current loss and possessing enhanced magnetic properties. A Fe/Mn_0.5_Zn_0.5_Fe_2_O_4_ SMC with inter-particle insulation was prepared by spark plasma sintering (SPS). The effect of the inter-particle insulation layer on the electro-magnetic properties of the Fe/Mn_0.5_Zn_0.5_Fe_2_O_4_ SMC was studied. It was demonstrated that the Fe/Mn_0.5_Zn_0.5_Fe_2_O_4_ composite compacts have excellent soft magnetic properties and low core loss and can be used to produce miniature magnetic components for applications in medium- and high-frequency fields. 

## 2. Experimental Materials and Methods

Pure Fe powder with a particle size of 25–45 μm (purity = 99.5%, Tianjin Metal Material Co., Ltd., Tianjin, China) and Mn_0.5_Zn_0.5_Fe_2_O_4_ powder with an average particle size of 30 nm (purity = 99.5%, Jingrui New Material Co., Ltd., Xuan Cheng, China) were used as raw materials at the mass ratio of 9:1. The specific experimental steps for preparing the coated iron powder and corresponding SMCs were as follows. (a) Fe/Mn_0.5_Zn_0.5_Fe_2_O_4_ mixed powder was premixed for 10 min, and then loaded into a high-energy ball mill (1-SL, Qingdao Meng Machinery Co., Ltd., Qingdao, China). The ball mill tank was protected by 0.3–0.5 MPa argon, and its outside was cooled by circulating water. The ball milling medium was a 3 mm small stainless steel ball, with the ball material ratio of 15:1, and the rotation speed of the ball mill was 200 r/min. The milling time was set as 1, 2 and 4 h. (b) The composite powder as-prepared was annealed in an argon atmosphere at 600 °C for 2 h to eliminate the internal stress. (c) The composite powder was pre-pressed with a graphite mold (ISO-68, Shanghai Dongyang Carbon Co., Ltd., Shanghai, China) and sintered in an SPS sintering furnace (Dr. Sinter 2040, Sumitomocoal Mining Co., Ltd., Kyoto, Japan) to prepare SMCs. All sintered samples were cut by an electric spark cutting machine to obtain an annular magnetic powder core (outer diameter 22 mm, inner diameter 12 mm, thickness 5 mm) (Figure 1). A diagram of the SPS system is shown in Figure 2. To prevent bonding during demolding, we padded graphite paper between the mold and the powder. The sintering process was as follows: sintering temperature 600 °C–900°C, holding time 4 min, heating rate 60 °C/min and sintering pressure 50 MPa. (d) The magnetic particle core was annealed again at 600 °C for 2 h in an argon flow atmosphere.

The particle size distributions of raw iron powder and composite powder were measured by an MS2000G laser particle size analyzer (Malvern, England). Phases were identified by a DX-2007 X-ray diffractometer (XRD, Dandong Fangyuan Co., Ltd., Dandong., China) using Cu Ka radiation at 30 kV and 30 ma. The morphology and local chemical uniformity of the composite powder and the annular magnetic particle core were detected by a scanning electron microscope (FEI Nova Nano SEM 450, Lincoln, NE, USA) and an energy-dispersive spectrometer (EDS, Ultra, EDAX, Mahwah, NJ, USA). The density of the annular magnetic particle core was measured following the Archimedes principle and using ethanol as the immersion solution. Static magnetic properties of the powder were tested at room temperature by a vibrating sample magnetometer (VSM, Quantum Design, San Diego, CA, USA). Total core loss of the annular magnetic particle core was measured in the range of 200 mT by a soft magnetic AC measuring instrument (Mats-2010sa/500 K, Linkioin, Loudi, China) from 1 kHz to 100 kHz. Hysteresis loss of the annular magnetic particle core was measured by a DC B-H ring tracer (Mat-2010sd, Linkioin, Loudi, China).

## 3. Results and Discussion

### 3.1. Microstructure and Magnetic Properties of Fe/Mn_0.5_Zn_0.5_Fe_2_O_4_ Composite Powder

Figure 3 shows the XRD patterns of the original Fe powder and the Fe/Mn_0.5_Zn_0.5_Fe_2_O_4_ powder with the change in milling time. Clearly, the original Fe powder shows three characteristic peaks on the XRD spectrum at 2θ of 45.18° (110), 65.70° (200) and 83.22° (211), which are consistent with the body-centered cubic structure of α-Fe. The XRD patterns of the Fe/Mn_0.5_Zn_0.5_Fe_2_O_4_ powder reflect the combination of two characteristic peaks of Fe and Mn_0.5_Zn_0.5_Fe_2_O_4_ and the strength in the peak of Mn_0.5_Zn_0.5_Fe_2_O_4_ is intensified with the prolonging of the ball milling time, meaning that Mn_0.5_Zn_0.5_Fe_2_O_4_ nano-powder can continuously coat Fe powder by mechanical ball milling. Because of the very good shaping of Fe, the Fe powder is difficult to grind into a complete amorphous state, which requires much time and a large ball material ratio [13]. Therefore, the diffraction peaks corresponding to iron are obvious, but become wider and shorter with the extension of the ball milling time. Compared with pure Fe powder, the diffraction peak of the coated powder is higher. The diffraction peak corresponding to Fe has a high-level shift, indicating that atoms have entered and distorted the iron crystal lattices [14]. In addition, the XRD spectrum shows no impurity phase, indicating that the Fe/Mn_0.5_Zn_0.5_Fe_2_O_4_ powder preparation method by mechanical ball milling coating is very efficient and clean.

Figure 4 shows the particle size distributions of the Fe/Mn_0.5_Zn_0.5_Fe_2_O_4_ powder before and after ball milling coating. After 2 h of mechanical ball milling, the particle size distribution curves of both types of powders show a single peak state, which still indicates a good particle size distribution concentration, with the average particle size of 45 mm [15]. This means that the mechanical ball milling process of 2 h does not lead to the crushing or agglomeration of Fe powder.

We selected the coated powder with a ball milling time of 2 h and compared it with the SEM of pure Fe powder (Figure 5). After ball milling for 2 h, the manganese Mn_0.5_Zn_0.5_Fe_2_O_4_ nano-powder is evenly coated on the surface of the Fe micro-powder, and the shape is basically spherical and quasi-spherical. This consequence is mainly due to the collision between the grinding medium and powders and between powders [16]. In addition, there is almost no crack distribution on the coated powder surface. We believe that the existence of Mn_0.5_Zn_0.5_Fe_2_O_4_ nano-powder will reduce the probability of cold welding between Fe powders, so there is almost no agglomeration [17]. Mn_0.5_Zn_0.5_Fe_2_O_4_ nano-powder acts as a process control agent in wet mechanical alloying, which is a similar role as that of benzene [18,19]. MnZn(Fe_2_O_4_)_2_ nano-powder can be uniformly coated on the surface of Fe powder, forming a rough-surfaced iron powder without small cracks [20].

Figure 6 demonstrates the hysteresis loops of the original powder and the coated powder. The original powder and the coated powder both have high saturation magnetization (M_S_) and low coercivity, which are excellent soft magnetic properties. From the change in M_S_, the M_S_ of the coated powder decreases continuously with the extension of the grinding time (from 1 to 4 h), but the decrease range is not obvious. M_S_ is insensitive to the structure and mainly related to the composition of the SMCs and the properties and quantity of the ferromagnetic phase [21].

Figure 7 shows the change in coercivity of the coated powder before and after heat treatment. Clearly, the extension of the ball milling time leads to an increase in coercivity, while the coercivity of the coated powder decreases after heat treatment. On the one hand, during the grinding process, a part of the energy remains as residual stress in the material, resulting in a large number of lattice distortion and crystal defects, pinning the magnetic domain movement [22], so the coercivity increases with the extension of the ball milling time. On the other hand, the internal stress of the coated powder is released after heat treatment, which reduces the coercivity. Figure 8 shows the XRD pattern of Fe/Mn_0.5_Zn_0.5_Fe_2_O_4_ composite powders after annealing. It can be found that no phase transformation for the composite powders was identified during annealing process; the Mn_0.5_Zn_0.5_Fe_2_O_4_ phase still maintained a monoclinic structure.

### 3.2. Densification of the Composite Powders

The SPS parameters, especially the sintering temperature, greatly influence the density and resistivity of SMCs. The Fe powder and Fe/Mn_0.5_Zn_0.5_Fe_2_O_4_ powder after ball milling for 2 h and annealing were selected as representatives. Figure 9 shows the effect of sintering temperature on the density of the bulk samples obtained after sintering of the pure Fe powder and the ball-milled Fe/Mn_0.5_Zn_0.5_Fe_2_O_4_ powder. Clearly, the density of these samples increases almost linearly with the sintering temperature rising from 600 °C to 900 °C. Under the same SPS conditions, the increase is significantly greater than that of the Fe/Mn_0.5_Zn_0.5_Fe_2_O_4_ SMCs. This is because the insulating layer of the Fe/Mn_0.5_Zn_0.5_Fe_2_O_4_ SMCs hinders the diffusion and flow between Fe particles during SPS, which complicates the formation of sintering necks and affects the full deformation, compaction and densification of the Fe/Mn_0.5_Zn_0.5_Fe_2_O_4_ SMCs. 

To further explore the variation law of the microstructure of the block sample with the sintering temperature, we probed into the relationship between the cross-section SEM photos of the sintered block containing Fe/Mn_0.5_Zn_0.5_Fe_2_O_4_ and the sintering temperature (Figure 10). The gray spherical area is Fe powder, and the gray black area is the manganese zinc ferrite insulation layer (Figure 10). The insulating layer of Mn_0.5_Zn_0.5_Fe_2_O_4_, which was non-conductive, was observed in the backscattering mode. With the increase in sintering temperature, Fe particles changed from a circular to quasi-circular shape, and even a strip shape appeared at 900 °C. The surrounding gray black area gradually became larger and wider. No discharge occurred at the beginning of SPS. With the rise in sintering temperature, the number of loaders in Mn_0.5_Zn_0.5_Fe_2_O_4_ increased, and the interaction between powders was intensified under the action of the electric field. Research on the SPS of ceramic powders shows that the dielectric characteristics of Mn_0.5_Zn_0.5_Fe_2_O_4_ are the main factor for the formation of plasma [23]. These powders, with a high dielectric constant, have a higher charge density on the surface. A large amount of plasma was produced along with the sintering process and the increase in sintering temperature. Spark discharge occurred between powders. Under the action of the pulse current, partial discharge produced much Joule heat. Local melting happened at the contact area between particles, and serious plastic deformation of powders occurred under the action of external pressure [24]. When the sintering temperature was very high, some small Fe particles and Mn_0.5_Zn_0.5_Fe_2_O_4_ nanoparticles melted and agglomerated. Hence, Mn_0.5_Zn_0.5_Fe_2_O_4_ cladding was destroyed [25].

To better compare the microstructure of Fe/Mn_0.5_Zn_0.5_Fe_2_O_4_ SMCs, we studied the SEM photos of block samples prepared by SPS and the pure Fe original powder at 900 °C (Figure 11). There is almost no hole or porosity on the surface of the sample, which is very dense.

### 3.3. Magnetic Properties of the Fe/Mn_0.5_Zn_0.5_Fe_2_O_4_ SMCs

Figure 12 shows the hysteresis loops of the Fe/Mn_0.5_Zn_0.5_Fe_2_O_4_ SMCs at four temperatures. The curve shape shows the typical low coercivity and high permeability of SMCs. Data of M_S_ and coercivity (H_C_) are shown in Figure 13 after sorting. After 700 °C, the sintered sample shows approximately equal M_S_ (Figure 12), which indicates the small dependence of M_S_ on sintering temperature in this experiment and confirms no obvious structural transformation of the Fe/Mn_0.5_Zn_0.5_Fe_2_O_4_ SMCs during sintering. However, the coercivity is obviously dependent on the sintering temperature, as it increases slightly at 600 °C–800 °C, but decreases sharply at 900 °C. We believe that with the increase in sintering temperature within 600 °C–800 °C, the Mn_0.5_Zn_0.5_Fe_2_O_4_ insulating layer causes domain wall perforation, hinders domain wall displacement and raises the energy required to complete magnetization and demagnetization, so H_C_ increases [26]. Furthermore, Hc can increase when increasing the grain size (MnZn ferrite) as the sintering temperature increases. When the temperature reaches 900 °C, the Fe/Mn_0.5_Zn_0.5_Fe_2_O_4_ SMC is over-burned (Figure 10d), which destroys the Mn_0.5_Zn_0.5_Fe_2_O_4_ insulating layer and reduces H_C_.

Figure 14 shows the amplitude permeability of the Fe/Mn_0.5_Zn_0.5_Fe_2_O_4_ SMCs prepared at different sintering temperatures and different frequencies. The amplitude permeability of the SMCs is stable, indicating that the ferrite insulation coating intensifies the resistivity between particles and reduces the inter-particular eddy current and the demagnetization field formed by the eddy current [27]. At a low frequency, the amplitude permeability μ_a_ of the Fe/Mn_0.5_Zn_0.5_Fe_2_O_4_ SMCs is much lower than that of the Fe alloys because μ_a_ corresponds to the slope of the initial magnetization curve and the initial permeability μ_i_. The μ_i_ and internal stress σ in a material are inversely proportional to the impurity concentration β. In this study, due to the addition of Mn_0.5_Zn_0.5_Fe_2_O_4_, the impurity concentration β of the Fe/Mn_0.5_Zn_0.5_Fe_2_O_4_ SMCs is much lower than that of the Fe alloys.

Figure 15 shows the SEM of Fe/Mn_0.5_Zn_0.5_Fe_2_O_4_ SMCs sintered at 700 °C and the corresponding element surface scanning diagram. Both O and Mn elements are distributed around the Fe element, while the Zn element is distributed across the whole area, but in a very small amount. This is because the Zn element easily reacts with oxygen to form ZnO. The composition analysis shows that the Fe powder is coated with Mn_0.5_Zn_0.5_Fe_2_O_4_ nano-powder. Hence, the isolation of the Fe powder is realized, so as to insulate Fe powder from the high-resistivity Mn_0.5_Zn_0.5_Fe_2_O_4_ nano-powder.

Figure 16 shows the resistivity test results of the Fe/Mn_0.5_Zn_0.5_Fe_2_O_4_ SMC sintered blocks prepared at different sintering temperatures. The resistivity of pure Fe powder is 25.47 μΩ·cm. With the increase in sintering temperature, the resistivity of the sintered block decreases from 1298.65 μΩ·cm at 600 °C down to 274.3 μΩ·cm at 800 °C. The reason is that the increasing density of the sintered blocks and the decreasing number of holes or gaps result in an intensification of electron mobility. However, with a further temperature rise, the resistivity decreases sharply to 85.5% at 900 °C. We believe that the significant change in resistance is mainly related to the percolation transition from the insulator to the conductor [28]. The Mn_0.5_Zn_0.5_Fe_2_O_4_ insulating layer is damaged, and a current conduction path is formed between some Fe powders, resulting in a sharp reduction in the resistance of sintered block materials.

Figure 17 demonstrates the loss variation curve under different sintering temperatures. The core loss increases significantly with the increment in frequency. The lowest loss occurs at 700 °C, while the loss increases sharply at 900 °C. On the micro-morphology of the sintered sample, the insulating layer of the Fe/Mn_0.5_Zn_0.5_Fe_2_O_4_ SMCs is increasingly tightly coated at 600 °C–700 °C, and the eddy current radius is decreasing (Figure 10). Therefore, the loss decreases. When the temperature rises to 900 °C, the Fe/Mn_0.5_Zn_0.5_Fe_2_O_4_ SMC is in an over-burned state, the insulating layer is damaged, and the eddy current loss between and within particles increases, resulting in more core loss (Figure 10d).

## 4. Conclusions

SMCs with the micro-cell structure were prepared using the SPS sintering spherical atomized Fe powder coated with Mn_0.5_Zn_0.5_Fe_2_O_4_ nano-powder.

(1)Mn_0.5_Zn_0.5_Fe_2_O_4_-coated Fe powder, corresponding Fe powder and corresponding Fe/Mn_0.5_Zn_0.5_Fe_2_O_4_ SMCs were successfully prepared by mechanical ball milling and SPS.(2)The structure and magnetic properties of the composite powders prepared by different ball milling time periods were studied. The results show that Mn_0.5_Zn_0.5_Fe_2_O_4_ nano-powder can be uniformly coated on spherical iron powder by mechanical ball milling, but it will produce defects and stress. With the extension of ball milling time, the coercivity increases, but after heat treatment, the coercivity can be reduced.(3)The Mn_0.5_Zn_0.5_Fe_2_O_4_ insulating layer isolates conductive iron particles and improves the resistivity of the magnetic particle core. With the increase in sintering temperature, the resistivity of the sintered block decreases from 1298.65 μΩ·cm down to 274.3 μΩ·cm, and shows the lowest core loss at 700 °C. Moreover, the amplitude permeability of Fe/Mn_0.5_Zn_0.5_Fe_2_O_4_ SMCs shows stability. Therefore, the Fe/Mn_0.5_Zn_0.5_Fe_2_O_4_ SMCs shows lower core loss and better magnetic properties, which make it possible to realize high energy conversion efficiency.

## Figures and Tables

**Figure 1 materials-15-01859-f001:**
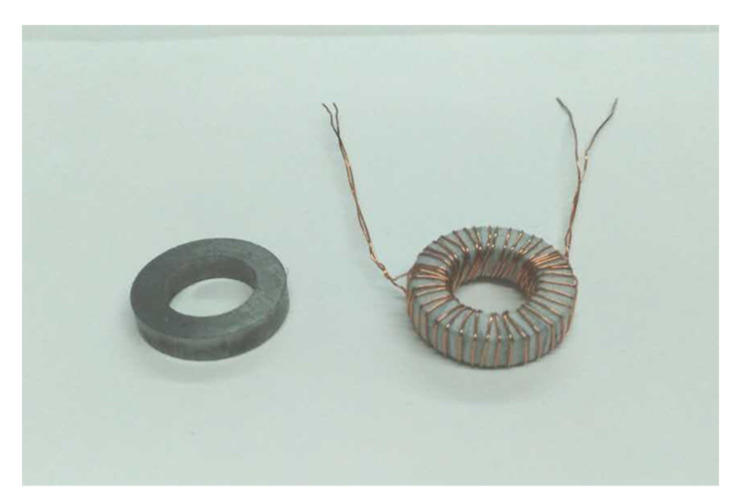
The toroidal-shaped magnetic cores of the Fe/Mn_0.5_Zn_0.5_Fe_2_O_4_ SMCs.

**Figure 2 materials-15-01859-f002:**
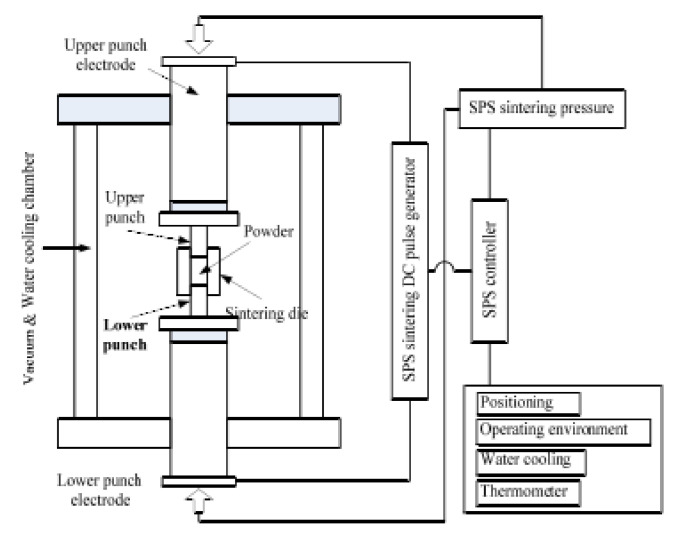
Diagram of SPS system for preparing SMCs in this work.

**Figure 3 materials-15-01859-f003:**
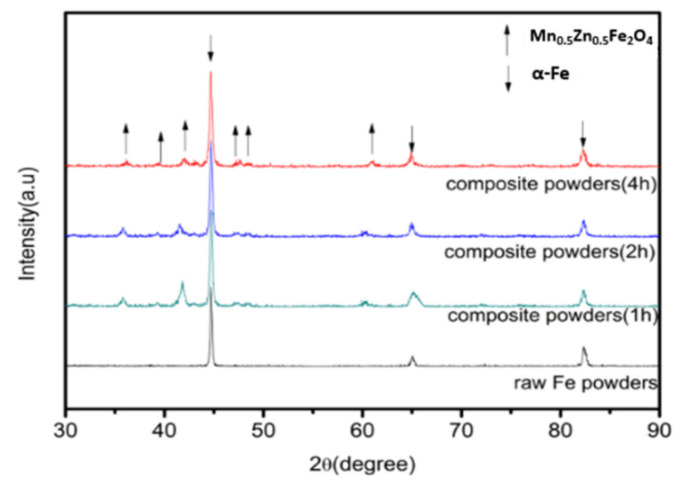
XRD pattern of Fe/Mn_0.5_Zn_0.5_Fe_2_O_4_ composite powders before and after encapsulation.

**Figure 4 materials-15-01859-f004:**
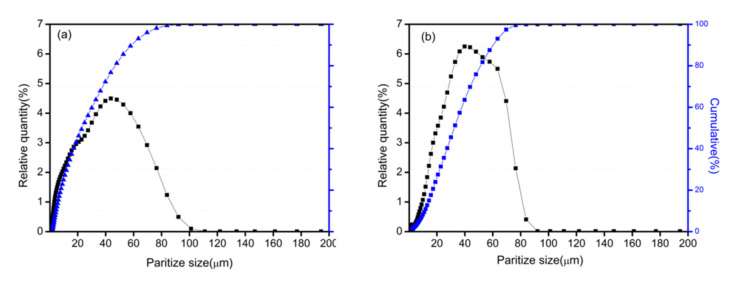
Particle size distribution of primary Fe powder and Fe/Mn_0.5_Zn_0.5_Fe_2_O_4_ core–shell structure powder (ball milled for 2h, 10%); (**a**) original Fe powder; (**b**) Fe/Mn_0.5_Zn_0.5_Fe_2_O_4_ powder.

**Figure 5 materials-15-01859-f005:**
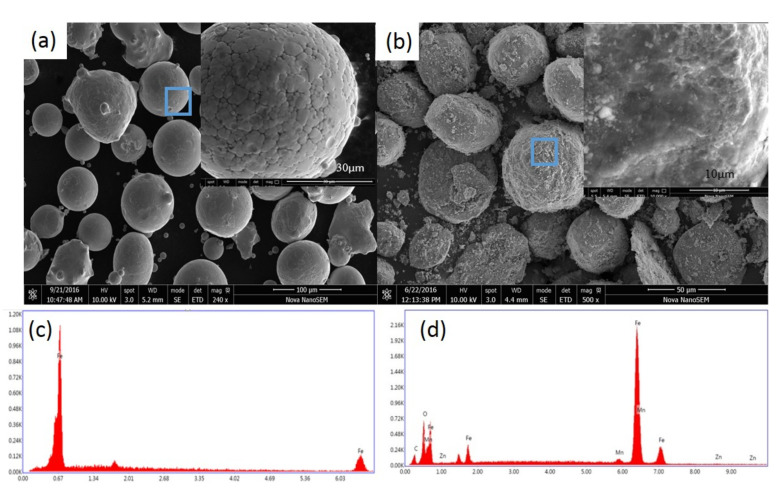
SEM image of primary Fe powder and Fe/Mn_0.5_Zn_0.5_Fe_2_O_4_ core–shell structure powder (ball-milled for 2 h, 10%); (**a**) original Fe powder; (**b**) EDS spectrum (**a**) of the selected region; (**c**) Fe/Mn_0.5_Zn_0.5_Fe_2_O_4_ core–shell structure powder; (**d**) EDS spectrum (**c**) of the selected region.

**Figure 6 materials-15-01859-f006:**
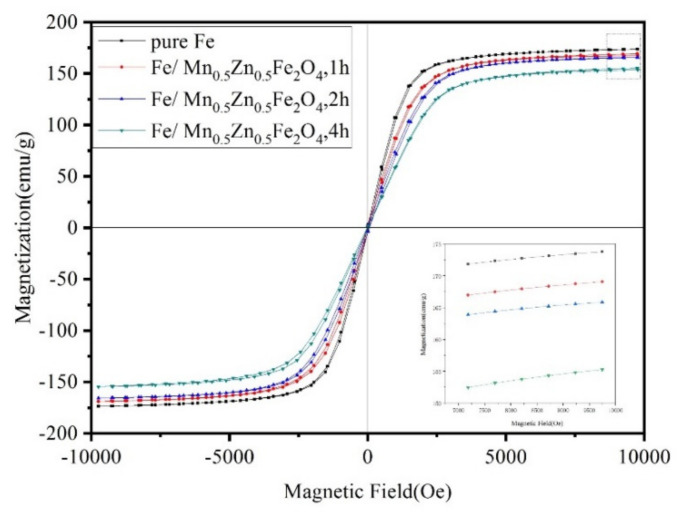
Hysteresis loop of original powder and Fe/Mn_0.5_Zn_0.5_Fe_2_O_4_ core–shell structure powder.

**Figure 7 materials-15-01859-f007:**
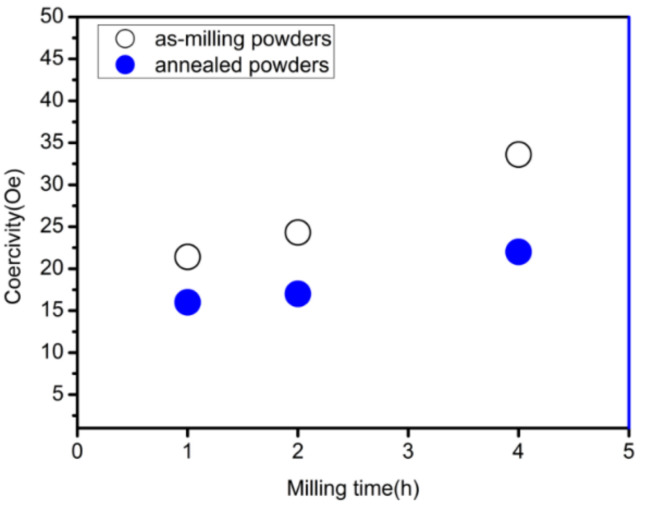
Coercive force before and after heat treatment of Fe/Mn_0.5_Zn_0.5_Fe_2_O_4_ core–shell powders with different milling times.

**Figure 8 materials-15-01859-f008:**
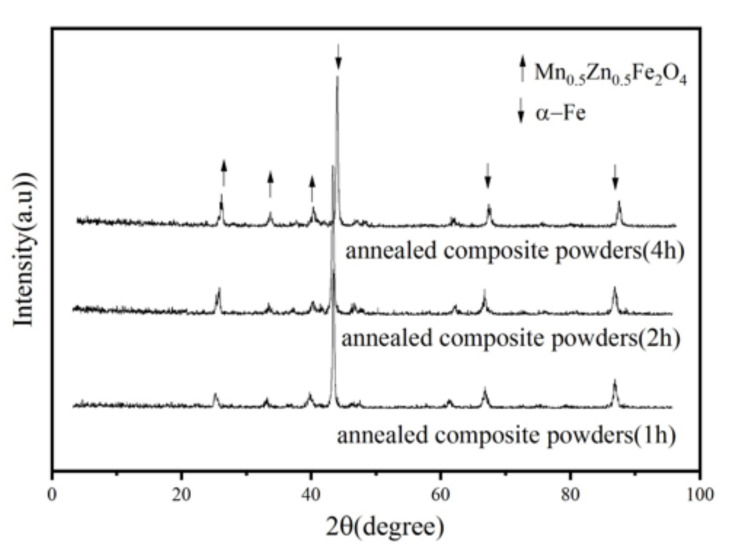
XRD pattern of Fe/Mn_0.5_Zn_0.5_Fe_2_O_4_ composite powders after annealing.

**Figure 9 materials-15-01859-f009:**
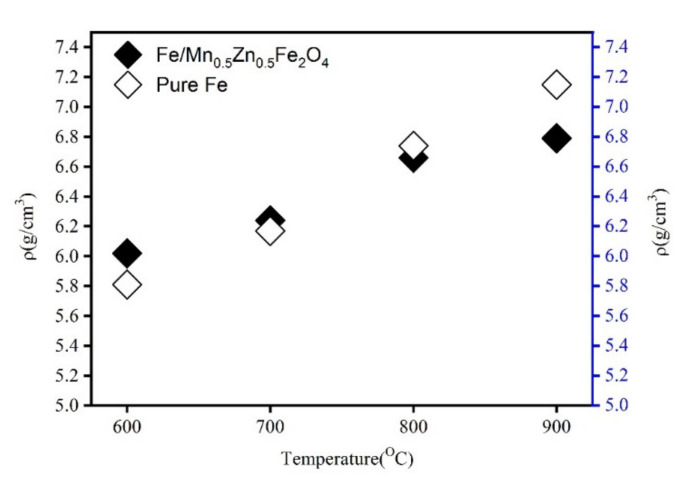
Density of pure Fe and Fe/Mn_0.5_Zn_0.5_Fe_2_O_4_ SMCs at different sintering temperatures.

**Figure 10 materials-15-01859-f010:**
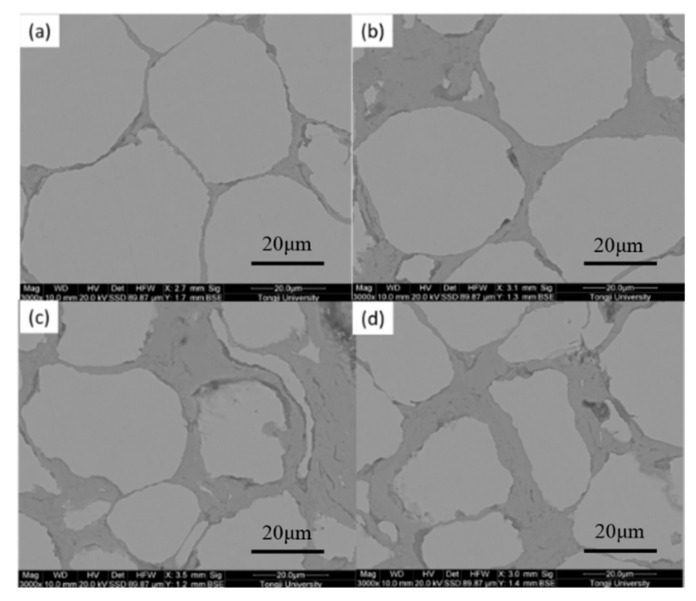
SEM images of Fe/Mn_0.5_Zn_0.5_Fe_2_O_4_ SMCs at different sintering temperatures (unetched). (**a**) 600 °C, (**b**) 700 °C, (**c**) 800 °C, (**d**) 900 °C.

**Figure 11 materials-15-01859-f011:**
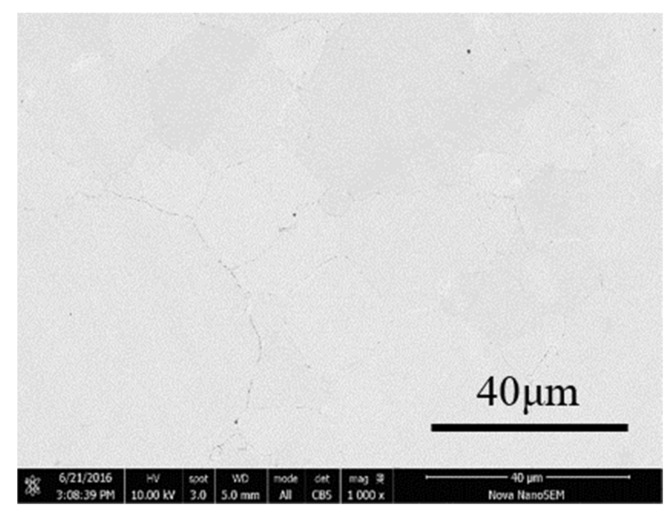
SEM image of original pure Fe powder bulk sample sintered at 900 °C.

**Figure 12 materials-15-01859-f012:**
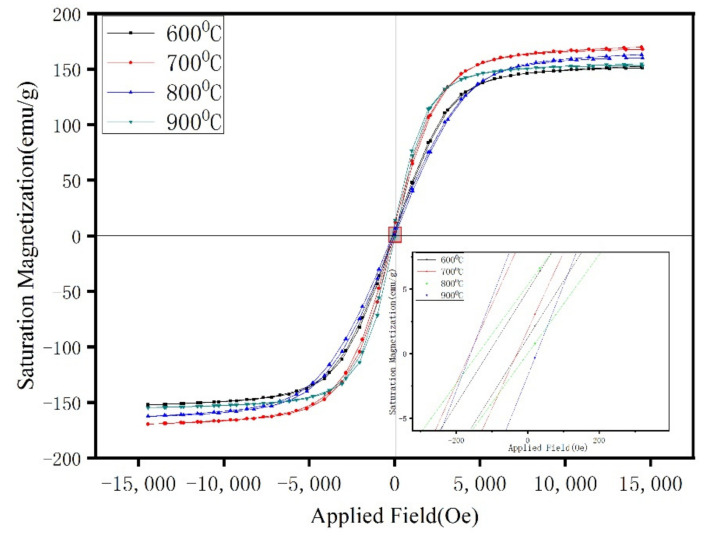
Hysteresis loop of Fe/Mn_0.5_Zn_0.5_Fe_2_O_4_ SMCs at different sintering temperatures.

**Figure 13 materials-15-01859-f013:**
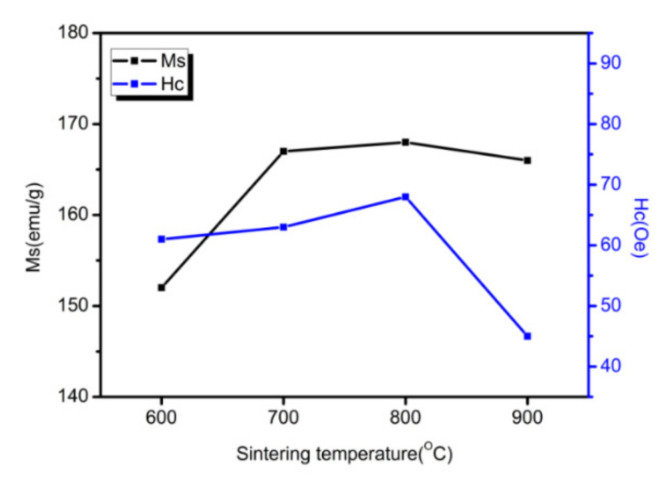
Saturation magnetization (Ms) and coercive force (Hc) of Fe/Mn_0.5_Zn_0.5_Fe_2_O_4_ SMCs at different sintering temperatures.

**Figure 14 materials-15-01859-f014:**
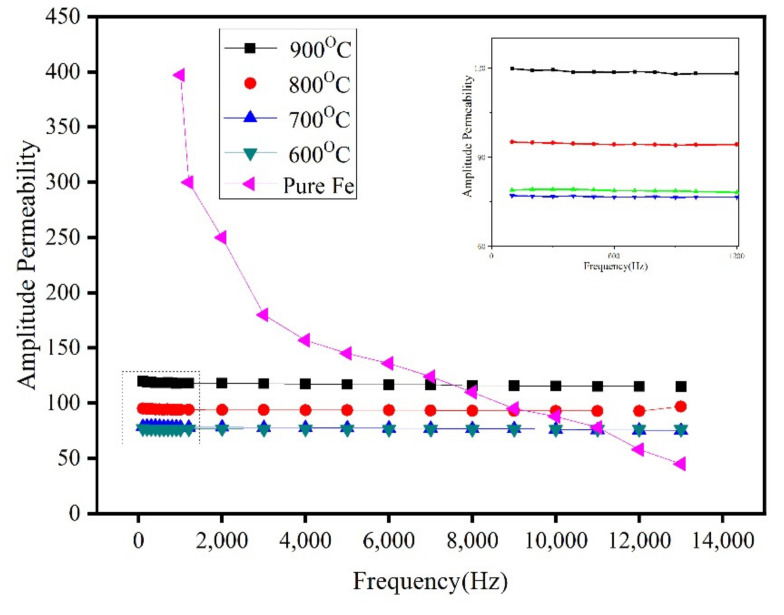
Amplitude permeability of Fe/Mn_0.5_Zn_0.5_Fe_2_O_4_ SMCs prepared at different sintering temperatures at different frequencies.

**Figure 15 materials-15-01859-f015:**
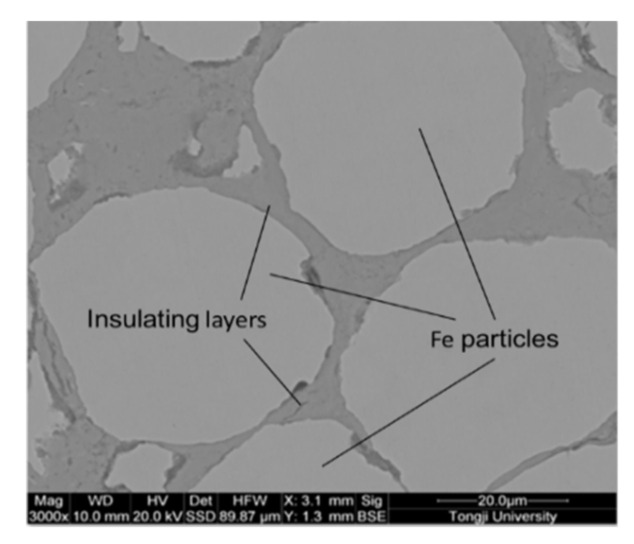
EDS spectrum of the polished surface for the Fe/Mn_0.5_Zn_0.5_Fe_2_O_4_ SMCs sintered at 700 °C.

**Figure 16 materials-15-01859-f016:**
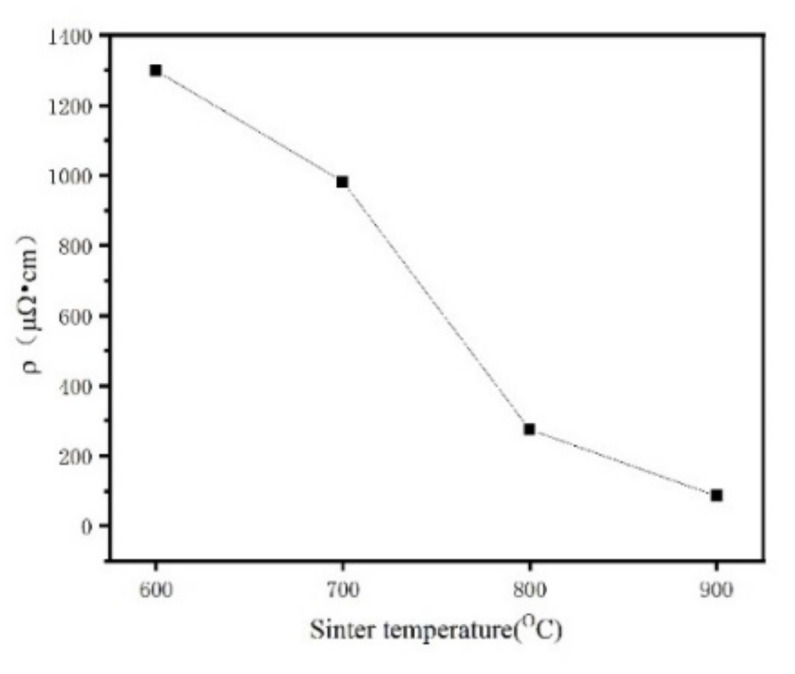
Influence of sintering temperature on the resistivity of the Fe/Mn_0.5_Zn_0.5_Fe_2_O_4_ SMCs prepared by SPS.

**Figure 17 materials-15-01859-f017:**
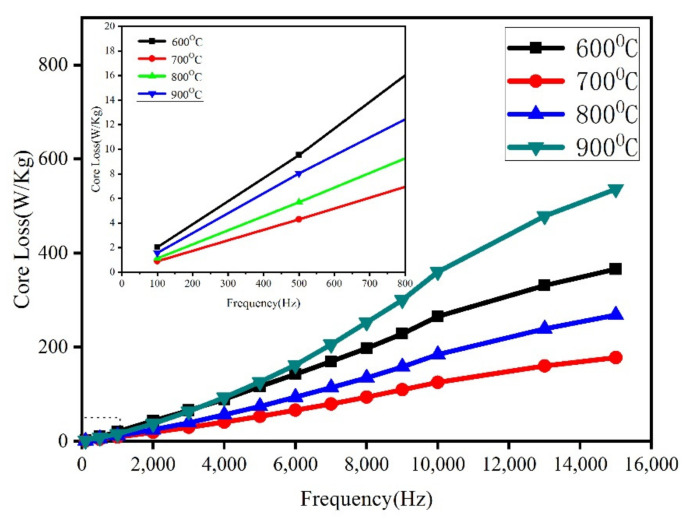
Core loss of Fe/Mn_0.5_Zn_0.5_Fe_2_O_4_ SMCs as a function of sintering temperature (Bm = 200 mT).

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
