# Peer review of "Microstructure and Magnetic Properties of Grain Boundary Insulated Fe/Mn0.5Zn0.5Fe2O4 Soft Magnetic Composites"

_materials, 2022, doi:10.3390/ma15051859_

Round 1

Reviewer 1 Report

The manuscript is written in poor English, is difficult to be read and needs substantial improvement in order to be considered for publication in any journal. I will not recommend the manuscript publication in Materials journal due to the following reasons:

  • The manuscript let me the impression of a manuscript written in a hurry! Many typos were identified: Mpa, 3-mm, 0.02t, 45 mm (particle size), μA, μI…..
  • The correct chemical formula for manganese zinc ferrite is MnZnFe2O4 and not MnZn(Fe2O4)2 as written in the document. What is the stoichiometry of the used ferrite, what is the ratio between Mn and Zn atoms (Mn1-xZnx Fe2O4)?
  • XRD investigations after annealing the milled powders is required to investigate if no reaction took place between Fe and ferrite particles!!!
  • The authors mentioned that they obtained core-shell particles and the ferrite is evenly distributed on the surface of the Fe particle. However, no evidence of such a particle/structure is given. Also, they mention that the obtained particles are spherical. I never obtained spherical particles by milling and, to the best of my knowledge, no paper reports such a result.
  • SEM analyses give little information to the readers in this case. EDX analysis on the cross-section of the particle is required.
  • “When the temperature exceeds 800℃, the Fe alloy density suddenly in-creases.” This is not visible in figure 8. A linear increase in the density is noticed.
  • The amount of ferrite contained by the sintered samples seems to be very different from sample to sample if we analyze figure 9.
  • Comment to figure 12: If the density of the sample increase upon increasing the sintering temperature, why the saturation magnetisation does not follow the same trend?
  • The EDX analysis presented in figure 14 needs to be revised. The distribution map of Zn is completely black. The map distribution of Mn is a mix map not a map distribution of a single element.
  • “With the increase of sintering temperature, the resistivity of the sintered block increases from 1298.65 μΩ•cm at 600℃ down to 274.3 μΩ•cm at 800℃.” In my opinion, the resistivity decreases upon increasing the sintering temperature (see figure 15)
  • The core loss was measured at 0.02 t (as mentioned in the experimental section) or at 100 mT as mentioned in the legend of figure 16?
  • The sintered samples were subjected to annealing at 600 °C for 2 hours. Why? It makes no sense to anneal a sintered sample.

After analysing the manuscript, I`m not sure what type of powders was used by the authors: Fe powder, Fe-Si powder or amorphous Fe-Si-B powders. In the title and the main part of the manuscript, the authors say they used Fe powder, in page 4 they say that Fe-Si powder was used (“nano-powder is evenly coated on the surface of iron silicon micro-powder”) and in the conclusion section is mentioned that amorphous Fe-Si-B was used. Moreover, this amorphous powder was coated with SiO2 : “SMCs with the micro-cell structure were prepared using the SPS sintering spherical atomized Fe-Si-B amorphous powder coated with SiO2 nanopowder.”

Author Response

1:The manuscript let me the impression of a manuscript written in a hurry! Many typos were identified: Mpa, 3-mm, 0.02t, 45 mm (particle size), μA, μI…..

It has been carefully revised and marked blue in the manuscript;

2:The correct chemical formula for manganese zinc ferrite is MnZnFe2O4 and not MnZn(Fe2O4)2 as written in the document. What is the stoichiometry of the used ferrite, what is the ratio between Mn and Zn atoms (Mn1-xZnx Fe2O4)?

MnZn(Fe2O4)2 modified to Mn0.5Zn0.5 Fe2O4  ;

3.XRD investigations after annealing the milled powders is required to investigate if no reaction took place between Fe and ferrite particles!!!

Add XRD investigations after annealing the milled powders.and found no reaction took place between Fe and ferrite particles.

4:The authors mentioned that they obtained core-shell particles and the ferrite is evenly distributed on the surface of the Fe particle. However, no evidence of such a particle/structure is given. Also, they mention that the obtained particles are spherical. I never obtained spherical particles by milling and, to the best of my knowledge, no paper reports such a result.

In order to prove this particle / structure, I made EDS on the surface of non ferrite particles and composite particles. In addition, spherical particles are obtained by grinding. You can refer to the following literature:

[1] Wu, Z. ,  Fan, X. ,  Jian, W. ,  Li, G. , &  Zhang, Z. . (2014). Core loss reduction in fe–6.5 wt.%si/sio2 core–shell composites by ball milling coating and spark plasma sintering. Journal of Alloys & Compounds, 617, 21-28.

[2]Z Luo, X Fan, W Hu, F Luo, G Li, & Y Liç­‰. (2019). Controllable sio2 insulating layer and magnetic properties for intergranular insulating fe-6.5wt.%si/sio2 composites. ADVANCED POWDER TECHNOLOGY.

5:SEM analyses give little information to the readers in this case. EDX analysis on the cross-section of the particle is required.

EDX analysis of particle cross section to prove that the core-shell structure is correct, but I can't complete such an experiment. Therefore, in order to prove the core-shell structure, I use another expression, that is, EDS analysis of the particle surface before and after coating, as shown in Figure 5

6:“When the temperature exceeds 800℃, the Fe alloy density suddenly in-creases.” This is not visible in figure 8. A linear increase in the density is noticed.

“When the temperature exceeds 800℃, the Fe alloy density suddenly in-creases.” is Deleted

7:Comment to figure 12: If the density of the sample increase upon increasing the sintering temperature, why the saturation magnetisation does not follow the same trend?

Since the saturation magnetization is not sensitive to the structure and structure, it is only related to the ferromagnetic phase

8:The EDX analysis presented in figure 14 needs to be revised. The distribution map of Zn is completely black. The map distribution of Mn is a mix map not a map distribution of a single element.

The EDX analysis presented in figure 14 is revised.

9:“With the increase of sintering temperature, the resistivity of the sintered block increases from 1298.65 μΩ•cm at 600℃ down to 274.3 μΩ•cm at 800℃.” In my opinion, the resistivity decreases upon increasing the sintering temperature (see figure 15)

“With the increase of sintering temperature, the resistivity of the sintered block increases from 1298.65 μΩ•cm at 600℃ down to 274.3 μΩ•cm at 800℃.”modified to :With the increase of sintering temperature,the resistivity of the sintered block decreases from 1298.65 μΩ•cm at 600℃ down to 274.3 μΩ•cm at 800℃.

10:The core loss was measured at 0.02 t (as mentioned in the experimental section) or at 100 mT as mentioned in the legend of figure 16?

The core loss was measured at 0.02 t ï¼›(Bm=100mT) modified to (Bm=200mT)

 11:  The sintered samples were subjected to annealing at 600 °C for 2 hours. Why? It makes no sense to anneal a sintered sample.

To removing residual stresses by annealing at 600 °C for 2 h under apure argon atmosphere.

12:After analysing the manuscript, I`m not sure what type of powders was used by the authors: Fe powder, Fe-Si powder or amorphous Fe-Si-B powders. In the title and the main part of the manuscript, the authors say they used Fe powder, in page 4 they say that Fe-Si powder was used (“nano-powder is evenly coated on the surface of iron silicon micro-powder”) and in the conclusion section is mentioned that amorphous Fe-Si-B was used. Moreover, this amorphous powder was coated with SiO2 : “SMCs with the micro-cell structure were prepared using the SPS sintering spherical atomized Fe-Si-B amorphous powder coated with SiO2 nanopowder.”

It has been carefully revised and marked blue in the manuscript

Reviewer 2 Report

This paper analyzes crystal structure, microstructure, and magnetic properties in Fe-MnZnFe2O4 composites.
The results provide valuable and interesting data to researchers who study soft magnetic composites and their applications.
The work is interesting and has potential for future research.

However, some explanation and analysis about the results are not enough.
Some of my comments that needs to check are below.

1. Fe/MnZnFe2O4 core-shell structure cannot be observed in Fig 5.
   Further, we cannot figure out the state of diffusion and embedding the MnZn ferrite.

2. We cannot distinguish simple mixtures and core-shell structures (MnZn ferrite-coated Fe) from M-H curve of Fig.6. 
   The Ms and Hc can increase in a core-shell structure because of the exchange effect.

3. The temperature should be shown in Fig. 9.

4. Hc can increase with increasing the grain size (MnZn ferrite) as the sintering temperature increase.

Author Response

  1. Fe/MnZnFe2O4 core-shell structure cannot be observed in Fig 5.

   Further, we cannot figure out the state of diffusion and embedding the MnZn ferrite.

   In order to characterize the core-shell structure, I replaced Figure 5, added EDS on the surface of Fe powder and coated composite powder, and “MnZn(Fe2O4)2 nano-powder can be gradually embedded into the surface cracks of iron powder, forming rough-surfaced iron powder without small cracks”modified the expression as follows: “MnZn(Fe2O4)2 nano-powder can be uniformly coated on the surface of iron powder, form-ing rough-surfaced iron powder without small cracks”

  1. We cannot distinguish simple mixtures and core-shell structures (MnZn ferrite-coated Fe) from M-H curve of Fig.6.

Figure 6 is redone and Fe / MnZnFe2O4 is marked.The Ms and Hc can increase in a core-shell structure because of the exchange effect.

  1. The temperature should be shown in Fig. 9.

Figure 9 SEM images of Fe/ MnZn(Fe2O4)2 SMCs at different sintering temperatures (unetched): (a)600℃(b)700℃(c)800℃(d)900℃

  1. Hc can increase with increasing the grain size (MnZn ferrite) as the sintering temperature increase.

Yes . we add : Whatmore , Hc can increase with increasing the grain size (MnZn ferrite) as the sintering temperature increase.

Reviewer 3 Report

The present research is focuse on evaluation of microstructure and magnetic properties of grain boundary insulated Fe / MnZn(Fe2O4)2 soft magnetic composites. 

Some minor remarks: 

  • Please check author's name...
  • last phrase from introduction should showcase the aim of the study with expected results
  • "Pure Fe powder in particle size of 45-25 μm" ...  should be written from small to big. 
  • Enlarge figure 2
  • Enlarge figure 3 and indicate the compounds on peaks
  • On fig. 5 and 9 - could you identify specific particles if Fe separate from the others to see the homogeneity 
  • What to see on figure 10 ?
  • Lack of recent literature from non-chinese researcher. 
  • Conclusions should be redrawn.. not to mention what was done.. but to outline the specific conclusions.... 
    • Mechanical ball milling can make MnZn(Fe2O4)2 nanopowder uniformly coat the spherical iron powder, but induce defects
      and stress. .. this remark can be found in literature... can be more specific...

Author Response

1.Please check author's name...

Modified to:Yan Liang 1    Yan Biao2   Peng lei 3,*

2.last phrase from introduction should showcase the aim of the study with expected results

Modified to: found these composite powders have homogeneous MnZn(Fe2O4)2 layers which are capa-ble of diminishing eddy-current loss, and possessing enhanced magnetic properties.

It is demonstrated that the Fe / MnZn(Fe2O4)2 composite compacts have excellent soft magnetic properties and low core loss. and can be used to produce miniature magnetic components for applications in medium and high frequency fields.

3."Pure Fe powder in particle size of 45-25 μm" ...  should be written from small to big.

Modified to:25-45 μm

4.On fig. 5 and 9 - could you identify specific particles if Fe separate from the others to see the homogeneity

Change the clearer Figure 5, reduce the multiple, and you can see the uniformity of particles and add EDS

5.What to see on figure 10 ?

The particle boundary can be seen by changing the clearer Figure 10

    6.Conclusions should be redrawn.. not to mention what was done.. but to outline the specific conclusions....

        Mechanical ball milling can make MnZn(Fe2O4)2 nanopowder uniformly coat the spherical iron powder, but induce defects

        and stress. .. this remark can be found in literature... can be more specific...

Conclusions Modified to: (1) MnZn(Fe2O4)2 coated Fe powder and corresponding Fe / MnZn(Fe2O4)2 SMCs were successfully prepared by mechanical ball milling and SPS.

(2) The structure and magnetic properties of the composite powders prepared by different ball milling time periods were studied. The results show that MnZn(Fe2O4)2 nano powder can be uniformly coated on spherical iron powder by mechanical ball milling, but it will produce defects and stress. With the extension of ball milling time, the coercivity increases, but after heat treatment, the coercivity can be reduced.

 (3) The MnZn(Fe2O4)2 insulating layer isolates conductive iron particles and improves the resistivity of the magnetic particle core. With the increase of sintering temperature, the resistivity of sintered block decreases from 1298.65μΩ•cm down to 274.3 μΩ•cm, and shows the lowest loss at 700 ℃. And the amplitude permeability of Fe / MnZn(Fe2O4)2 alloy shows stability. Therefore, the Fe / MnZn(Fe2O4)2 SMCs shows lower core loss and better magnetic properties, which make it possible to realize high energy conversion efficiency.

Round 2

Reviewer 1 Report

The manuscript was revised according to the reviewers suggestions.